# A ten-year review of indications and outcomes of obstetric admissions to an intensive care unit in a low-resource country

Betty Anane-Fenin[1]*, Evans Kofi Agbeno[2], Joseph Osarfo[3], Douglas Aninng Opoku Anning[4], Abigail Serwaa Boateng[1‡], Sebastian Ken-Amoah[2‡], Anthony Ofori Amanfo[2‡], Leonard Derkyi-Kwarteng[5‡], Mohammed Mouhajer[2‡], Sarah Ama Amoo[6‡], Joycelyn Ashong[1‡], Ernestina Jeffery[6‡]

1 Department of Obstetrics and Gynaecology, Cape Coast Teaching Hospital, Cape Coast, Ghana,
2 Department of Obstetrics and Gynaecology, University of Cape Coast School of Medical Sciences, Cape Coast, Ghana, 3 Department of Community Medicine, School of Medicine, University of Health and Allied Sciences, Ho, Ghana, 4 School of Public Health, Kwame Nkrumah University of Science and Technology, Kumasi, Ghana, 5 Department of Pathology, University of Cape Coast School of Medical Sciences, Cape Coast, Ghana, 6 Intensive Care Unit, Cape Coast Teaching Hospital, Cape Coast, Ghana

☯ These authors contributed equally to this work.
‡ ASB, SK-A, AOA, LD-K, MM, SAA, JA and EJ also contributed equally to this work.
* oparebea81@gmail.com

**Data Availability Statement:** All relevant data are within the manuscript and its Supporting Information files

## Abstract

### Introduction

Obstetric intensive care unit admission (ICU) suggests severe morbidity. However, there is no available data on the subject in Ghana. This retrospective review was conducted to determine the indications for obstetric ICU admission, their outcomes and factors influencing these outcomes to aid continuous quality improvement in obstetric care.

### Methods

This was a retrospective review conducted in a tertiary hospital in Ghana. Data on participant characteristics including age and whether participant was intubated were collected from patient records for all obstetric ICU admissions from 1st January 2010 to 31st December 2019. Descriptive statistics were presented as frequencies, proportions and charts. Hazard ratios were generated for relations between obstetric ICU admission outcome and participant characteristics. A p-value <0.05 was deemed statistically significant.

### Results

There were 443 obstetric ICU admissions over the review period making up 25.7% of all ICU admissions. The commonest indications for obstetric ICU admissions were hypertensive disorders of pregnancy (70.4%, n = 312/443), hemorrhage (14.4%, n = 64/443) and sepsis (9.3%, n = 41/443). The case fatality rates for hypertension, hemorrhage, and sepsis were 17.6%, 37.5%, and 63.4% respectively. The obstetric ICU mortality rate was 26% (115/443) over the review period. Age ≥25 years and a need for mechanical ventilation carried

**Funding:** The authors received no specific funding for this work.

**Competing interests:** The authors have declared that no competing interests exist.

increased mortality risks following ICU admission while surgery in the index pregnancy was associated with a reduced risk of death.

## Conclusion

Hypertension, haemorrhage and sepsis are the leading indications for obstetric ICU admissions. Thus, preeclampsia screening and prevention, as well as intensifying antenatal education on the danger signs of pregnancy can minimize obstetric complications. The establishment of an obstetric HDU in CCTH and the strengthening of communication between specialists and the healthcare providers in the lower facilities, are also essential for improved pregnancy outcomes. Further studies are needed to better appreciate the wider issues underlying obstetric ICU admission outcomes.

## Plain language summary

This was a review of the reasons for admitting severely-ill pregnant women and women who had delivered within the past 42 days to the intensive care unit (ICU), the admission outcomes and risk factors associated with ICU mortality in a tertiary hospital in a low-resource country. High blood pressure and its complications, bleeding and severe infections were observed as the three most significant reasons for ICU admissions in decreasing order of significance. Pre-existing medical conditions and those arising as a result of, or aggravated by pregnancy; obstructed labour and post-operative monitoring were the other reasons for ICU admission over the study period. Overall, 26% of the admitted patients died at the ICU and maternal age of at least 25 years and the need for intubation were identified as risk factors for ICU deaths. Attention must be paid to high blood pressure during pregnancy.

## Introduction

Pregnancy and the puerperium are risk factors for severe morbidity and mortality in women in their reproductive age [1, 2]. The physiological and anatomical changes that occur during this period may also make managing maternal medical and other disease conditions very challenging [1]. The occurrence of morbidities that are unique to this period, and the effect of disease and medication on both mother and fetus/baby are key considerations in the management of the obstetric patient.

With the decline in maternal mortalities worldwide [3], evaluation of maternal near-misses, also known as severe acute maternal morbidity (SAMM), which is defined as *"a woman who nearly died but survived a complication that occurred during pregnancy, childbirth or within 42 days of termination of pregnancy"* [4], has become a helpful tool for the assessment of the quality of obstetric care. Women who experience life-threatening complications, either as a near-miss or mortality, are more likely to be admitted to the intensive care unit (ICU), where a multidisciplinary approach to their care can optimize outcomes. The tool developed by World Health Organization (WHO) for the evaluation of near-misses defines maternal near-miss based on three criteria–disease criteria, intervention criteria, and organ dysfunction-based criteria [5]. Admission to the intensive care unit falls under the intervention criteria and is defined as *"admission to a unit that provides 24-hour medical supervision and is able to provide mechanical ventilation and continuous vasoactive drug support"* [4]. These admissions are considered an objective indicator of severe maternal morbidity.

The incidence of obstetric ICU admissions is generally reported as <1%, but a systematic review by Pollock and colleagues revealed rates up to 16%, accounting for a median of about 2.7 per 1000 deliveries [1, 6]. Fewer admissions are observed in high income countries than in the low-resourced where about half of the global maternal deaths occur [2]. The greater number of admissions is compounded by a general lack of ICU beds in low-resource countries and most of those available are located in the large referral hospitals in the cities [7]. There is thus a deficit in the care of critically ill patients including obstetric cases. This situation is not different in Ghana with only 149 beds in 16 functioning ICUs for a population of about 30 million as of 2020 [8].

Women in their puerperium are more at risk of ICU admission than pregnant women [1, 9]. The incidence of severe disease, and the rates of mechanical ventilation and mortality are also observed to be higher in Sub-Saharan Africa than in developed countries [1]. As many as 90% of ICU patients in developing countries required mechanical ventilation [10], whereas much lower rates of 19%-45% have been reported in high-income countries such as Netherlands and Australia [11–13]. Mortality rates following obstetric ICU admission of 34.8%-49% have been reported in sub-Saharan Africa [2, 6, 10] compared to 5.1% in China and <1% in Australia and New Zealand [11, 14]. A mean duration of stay of 3–4 days has been reported in Asia and Europe [12, 15] and hypertensive disorders of pregnancy (HDP), hemorrhage, and sepsis are often the main indications for ICU admission [1, 2, 6].

The subject of indications and outcomes of obstetric ICU admissions has not been explored in Ghana. Furthermore, obstetric ICU admissions and their outcomes of survival/mortality are typically immediately known to the particular team of doctors directly responsible for the patient but remain oblivious to others at the departmental or even hospital level until much later when mortalities are reviewed. This is deemed a problem and it is pertinent to have an overarching knowledge of obstetric ICU admission indications, trends and outcomes that inure to the benefit of all stakeholders including clinicians, managers and those in the health planning and policy space.

This ten- year review was conducted to obtain an overview of indications for obstetric ICU admission, outcomes and factors influencing these outcomes. It is expected to provide evidence-based insights for much desired continuous quality improvement, especially for the critically ill, in the Cape Coast Teaching Hospital and other tertiary facilities in Ghana and sub-Saharan Africa.

## Materials and methods

### Study design, site and population

This was a retrospective descriptive study carried out in the Cape Coast Teaching Hospital (CCTH) in Ghana involving all obstetric patients (including those up to 42 days post-delivery), who were admitted at the ICU from 1st January 2010 to 31st December 2019.

The hospital is located in the Central Region and is the highest referral center for the Central and Western Regions of Ghana and parts of the Ashanti region. It is a 400-bed facility, of which 65 are dedicated to the Obstetrics and Gynaecology department, with a bed occupancy of 70%. Over three thousand (3000) deliveries are conducted annually with an average caesarean section rate of 39% (Biostatistics Unit, CCTH, 2018; unpublished).

Day-to-day obstetric and gynecological emergencies are received from other hospitals in our catchment area and managed at the Emergency Triaging and Treatment (ETAT) center of the department. Intrapartum emergencies among women delivering in CCTH are managed at the delivery suite. The department lacks a high-dependency unit (HDU) or high-care area (HCA). The entire hospital has a 5-bed capacity ICU; a level 1 ICU according to the World

Federation of Societies of Intensive and Critical Care Medicine (WFSICCM) Classification of ICUs [16], which caters for all cases requiring ICU attention. Obstetric cases are admitted to the ICU variously from the theatre, the Obstetrics and Gynaecology wards, ETAT center and the delivery suite. For emphasis, all obstetrics-related cases referred to the hospital are routed through the ETAT before being sent to the ICU if intensive care is needed. The unit is managed by one anesthesiologist/intensive care physician, nurse anaesthetists, medical officers, critical care and general nurses. The obstetricians also review the patients daily and make obstetric-specific inputs.

## Data collection

The review involved all obstetric ICU admissions from 2010 to 2019 and hence no sample size determination was required. The inclusion criterion was obstetric ICU admission (including those up to 42 days post-partum) for any period from 1st January 2010 to 31st December 2019. A participant was to be excluded if data could not be found on ≥40% of the planned variables.

From 3rd February, 2020 to 31st July, 2020, data on patient age, occupation, diagnosis, dates of admission and discharge, whether patient had a caesarean section/ hysterotomy, post-partum laparotomy for exploration, uterine repair or postpartum hysterectomy in the index pregnancy prior to ICU admission or required mechanical ventilation during admission at the ICU and the final outcome following admission were extracted mainly from archived scanned versions of patient folders, the admissions and discharges (A&D) register, the report and incident books at the ICU, the patient register at the theatre as well as patient records from the hospital's electronic data registry. This was done for all obstetric patients meeting the inclusion criterion and none was excluded. Data extraction was done by two residents in the department who were trained for two days on the study objectives, definition of terminologies and the study protocols. A template designed by the first author was used for the extraction. For quality control, at least 70% of the data collected was cross-checked against primary sources for accuracy and consistency by three obstetricians who worked closely with the residents. Data on other relevant demographics such as level of education, parity and gravidity were poorly documented and could not be used. To give more context, the hospital used paper patient folders until 2017. In 2018, patient records went fully electronic as part of a nationwide implementation of electronic health record-keeping based on the international classification of disease (ICD) coding system and patient data could be accessed at all points of care in the hospital. Hard-paper patient folders were scanned and archived but a 100% completion may not have been achieved. Data collection for the review thus employed both scanned folders and the electronic registry among others as earlier indicated. Also, all points of care keep summary records of patient care and these records were used to triangulate data collected and to fill in missing information as much as possible.

In addition, total ICU admissions and deaths (obstetric and non-obstetric), total number of deliveries and the total number of maternal mortalities recorded in the hospital (in and outside the ICU) over the review period were also captured to contextually situate obstetric ICU admissions and deaths.

## Description of diagnoses categories used

The categories of diagnoses used were arbitrary since the electronic record system with ICD codes was implemented only in 2018.

The category 'Uncomplicated Hypertension' was used to represent any hypertensive disorder of pregnancy that was not associated with any of the following severe features: organ dysfunction, HELLP syndrome (hemolysis, elevated liver enzymes, low platelets), eclampsia,

pulmonary edema, uncontrollable blood pressures, cerebrovascular accident (CVA), intrauterine fetal death (IUFD) with or without placental abruption, and placental abruption with or without disseminated intravascular coagulopathy (DIC). Hence, it encompassed chronic hypertension (hypertension that either predates pregnancy or diagnosed before 20 weeks of pregnancy), gestational hypertension (non-proteinuric hypertension diagnosed after 20 weeks of pregnancy) and preeclampsia without severe features (controlled proteinuric hypertension diagnosed from 20 weeks without organ dysfunction).

'*Complicated Hypertension*' refers to hypertension with any of the severe features mentioned above.

'*Hemorrhage*' represents both antepartum and postpartum hemorrhages, excluding DIC from complicated hypertension (which is captured under complicated hypertension).

The diagnosis of '*sepsis*' was used for patients with a systemic inflammatory response with organ damage from a suspected or confirmed infectious cause.

The combined categories, '*sepsis and uncomplicated hypertension*', '*sepsis and complicated hypertension*', '*sepsis and hemorrhage*' were used for patients, who had clinical evidence of both conditions.

The '*medical*' category comprised both preexisting medical conditions such as diabetes, asthma, sickle cell disease, and also medical conditions that occurred whiles pregnant or up to 42 days post-delivery, such as pneumonia, pulmonary embolism, and peripartum cardiomyopathy.

## Data analysis

Data was double-entered in Excel (Microsoft, USA), cleaned and exported into Stata 14 (College Station, TX, USA) for analysis. Descriptive statistics were presented as frequencies, proportions, percentages, mean with standard deviation and median with range and charts. The outcome variable was the status of the patient upon exiting from the ICU (dead or alive) and Cox regression analysis was employed to generate hazard ratios for relations with the participants' background and clinical characteristics such as age and whether or not surgery was done. The significance level was set at 5%, and 95% confidence intervals are reported. Variables with significant p-values ($p < 0.05$ at 95% confidence interval) in univariate analysis were entered into a multivariable regression model for adjusted hazard ratios. There was only one group of patients in consideration here but Kaplan-Meier survival graphs were generated for comparison of survival among the different categories of the variables; age, surgery and intubation.

## Ethical considerations

Ethical clearance for this secondary data review was granted by the Ethical Review Committee of CCTH with reference number CCTHERC/EC/2020/080. All data were fully anonymized using study identification numbers to assure confidentiality. The study was retrospective in approach and spanned a decade. Informed consent was not obtained as participants had either been long discharged or dead at the time of data extraction. This was made clear in the study proposal submitted for ethical review and it did not generate any queries. The hospital management also gave written permission to use patient data.

## Results

### Background and clinical characteristics of study participants

Four hundred and forty-three (443) obstetric patients were admitted to the ICU over the defined study period and they all contributed data to the review. None of the participants had

a repeat ICU admission on a different pregnancy. Each participant thus contributed data only once to the review. Over the study period, 1721 patients were admitted to the ICU (total obstetric and non-obstetric) and 683 died.

These 443 obstetric ICU admissions constituted a quarter of all ICU admissions over the 10-year review period (25.7%, 443/1721). With a total of 30, 203 deliveries, there were 14.7 obstetric ICU admissions per 1000 deliveries. About a quarter of the obstetric ICU admissions died (26%, 115/443) and this number made up nearly two-fifths (38.6%, 115/298) of total maternal mortalities in the hospital over the review period as well as 16.8% (115/683) of total ICU deaths (obstetric plus non-obstetric).

The background and clinical characteristics of the participants are presented in Table 1. Their ages ranged from 14 to 48 years with a mean (SD) of 27.4 years (7.1). Majority of the participants worked in the informal sector (64.9%, 268/443). About 47% (207/443) had had either a caesarean section, laparotomy for exploration, uterine repair or postpartum hysterectomy prior to ICU admission while 28.9% (128/443) had been intubated sometime during their ICU admission. The median duration of stay in the ICU was 2 days with a range of 1 to 45 days. More than three-quarters (334/443) of the participants stayed for 1–3 days. The mean duration of stay (SD) was 2.8 days (±3.2).

**Table 1. Background and clinical characteristics of study participants.**

| Variables | Frequency (N = 443) | Percentage (%) |
|---|---|---|
| **Age (years)** | | |
| ≤ 25 | 166 | 37.5 |
| 25–35 | 218 | 49.2 |
| >35 | 59 | 13.3 |
| Mean Age (±SD) | 27.4 (±7.1) | |
| [a]**Occupation** | | |
| [b]Formal | 40 | 9.7 |
| [c]Informal | 268 | 64.9 |
| Unemployed | 105 | 25.4 |
| **Duration of stay (days)** | | |
| 1–3 | 334 | 75.4 |
| 4–7 | 72 | 16.3 |
| ≥ 8 | 22 | 5.0 |
| Not indicated | 15 | 3.4 |
| Mean (±SD) | 2.8 (±3.2) | |
| **Surgery** | | |
| Yes | 207 | 46.7 |
| No | 236 | 53.3 |
| **Intubation** | | |
| Yes | 128 | 28.9 |
| No | 315 | 71.1 |
| **Outcome** | | |
| Transferred to the ward | 328 | 74.0 |
| Died | 115 | 26.0 |

[a]for occupation, N was 413

[b]formal occupation included public servants and corporate workers

[c]informal occupation included traders, caterers, farmers, artisans, etc

### Indications for ICU admissions

Majority of the participants were admitted on account of hypertension (70.4%, 312/443) with only one and three cases relating to obstructed labour and post-operative monitoring respectively (see Fig 1). Of the 312 participants that had HDP, 218 (69.9%) had complicated hypertension (see Fig 2).

### Description of trends in indications for obstetric ICU admission comparing 2010–2014 and 2015–2019

Hypertension and obstetric haemorrhage remained the top two indications for obstetric ICU admission over the ten-year review period. Hypertension accounted for over 80% (241/296) of obstetric ICU admissions over the first half of the review period (2010–2014) (see Fig 3). However, this decreased to 48.3% (71/147) over 2015–2019. The burden of obstetric haemorrhage essentially doubled from 11.1% (33/296) in the period 2010–2014 to 21.1% (31/147) over 2015–2019.

Medical causes and sepsis, as indications for obstetric ICU admission, increased in burden at least 5-fold and 3-fold respectively between 2010–2014 and 2015–2019. Obstetric ICU mortality rates also increased 2-fold from 18.6% (55/296) in 2010–2014 to 40.8% (60/147) in 2015–2019.

### Factors influencing outcomes following obstetric ICU admission

About 14.5% (62/428) of study participants died on day 1 of ICU admission, 4.4% (19/428) on day 2 and 2.3% (10/428) on day 3. More than half of the ICU obstetric deaths (53.9%, 62/115) occurred within 24 hours of admission, 16.5% (19/115) on day 2 and 8.7% (10/115) on day 3. The median survival time is 10 days (95% CI 6,14).

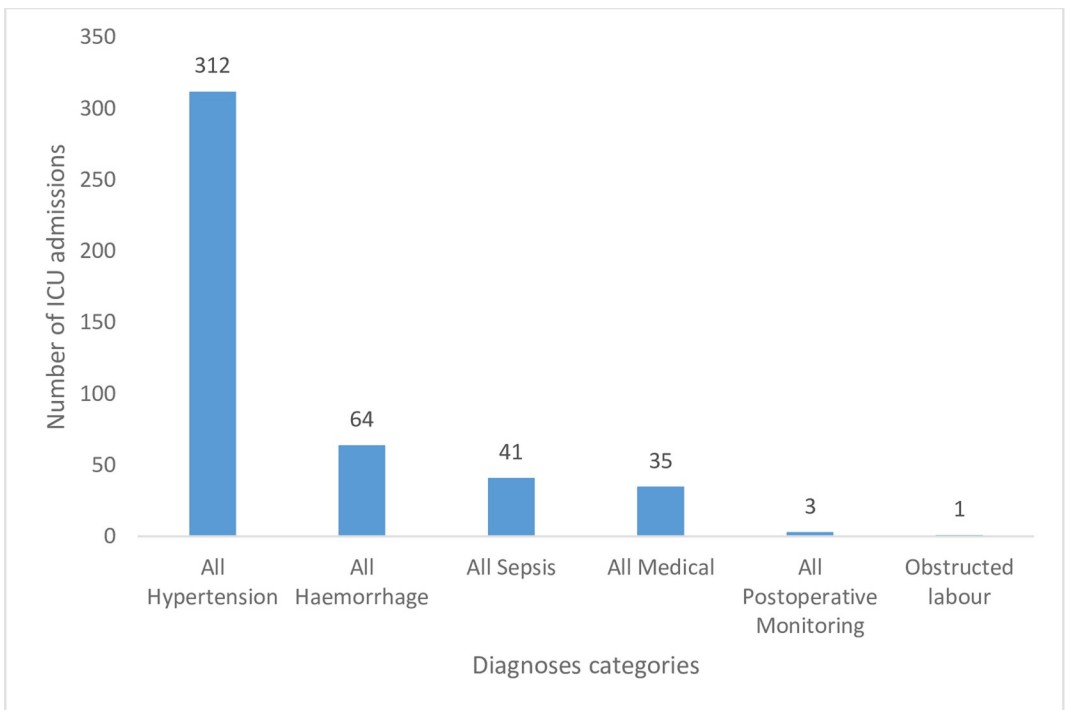

**Fig 1. Indications and numbers of obstetric ICU admissions in Cape Coast Teaching Hospital from 2010 to 2019 (the sum exceeds 443 as there were individuals with multiple indications).**

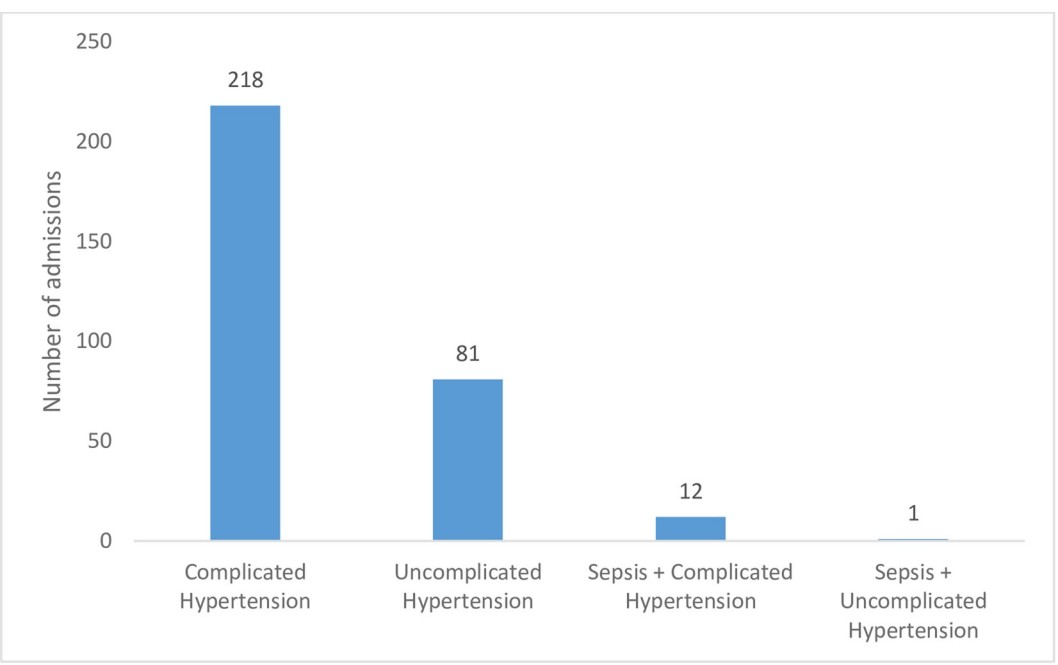

**Fig 2. Number of obstetric ICU admissions for various categories of hypertension among participants with hypertension.**

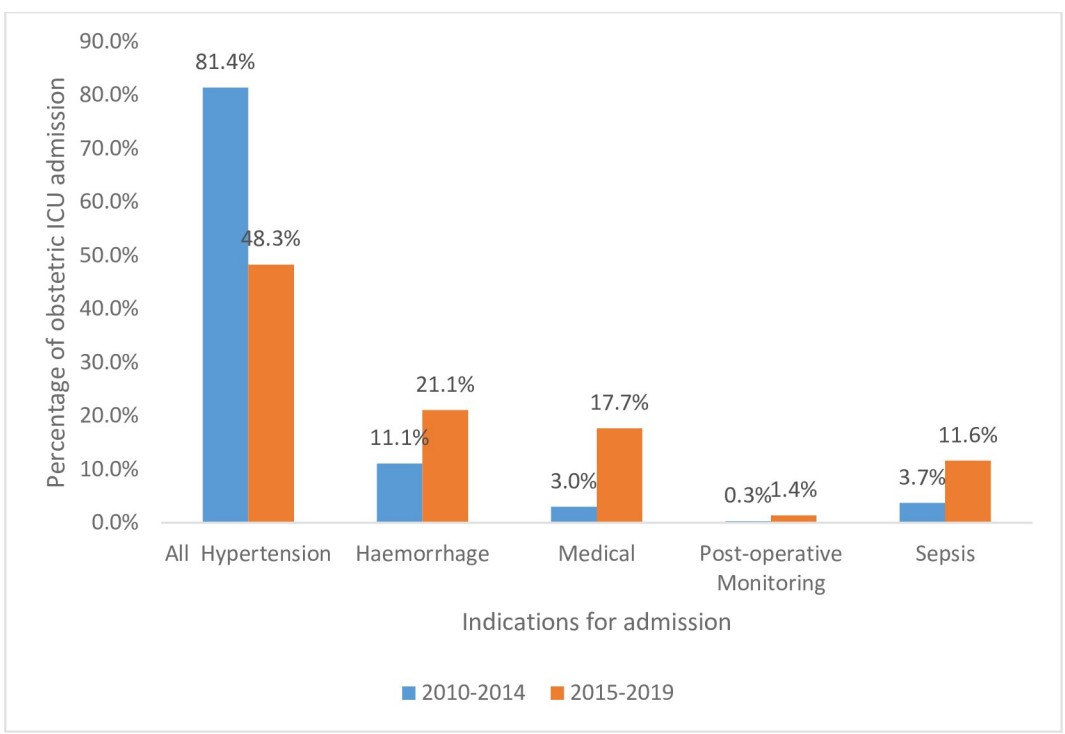

**Fig 3. Comparison of major indications for obstetric ICU admission in Cape Coast Teaching Hospital between 2010–2014 and 2015–2019.**

The variables age, surgery and intubation were significantly associated with death or survival outcomes following obstetric ICU admission in univariate Cox regression analysis (see Table 2). Compared to the youngest age group (<25 years), women aged above 35 years had twice the risk of death [HR 2.18, 95% CI 1.24, 3.84; p = 0.007] while those in the age group 25–35 years had a 67% increased risk of death [HR 1.67, 95% CI 1.06, 2.62; p = 0.025]. Those who were intubated were, at least, six times more at risk of death compared to those who were not. Those who had surgery had 66% of the risk in the group that did not have surgery [HR 0.66, 95%CI 0.45, 0.97; p = 0.034]. Participants' occupation did not influence obstetric ICU admission outcomes. In the multivariable analysis, intubation [AHR 6.37, 95% CI 4.23, 9.57; p<0.001] and age greater than 35 years [AHR 1.88, 95% CI 1.07, 3.31; p = 0.028] remained strongly associated with a death outcome.

Fig 4 shows Kaplan-Meier survival estimates to compare survival functions, over the review period, for the different groups based on whether there was surgery or not, intubation or not, the different categories of age, and the overall survival estimates. Testing for equality of survivor functions, there were significant differences in survival among the different age groups (p = 0.009), between those intubated and those not intubated (p < 0.001) and those who had surgery and those who did not (p = 0.024). Within the first 5 days, the survivor function was 0.7664 (95%CI 0.7233, 0.8036). In the second 5 days (i.e. days 5–9), survivor function was 0.6282 (95%CI 0.5415, 0.7030). In the third five days (i.e. 10–14), the survivor function was 0.4607 (95%CI 0.3047, 0.6032) and this particular survivor function remained constant over the next grouped days (15–45). Censoring time at day 15, the survivor functions for the first, second and third set of 5 days remained as reported above. However, the survivor function changed for the last set of days (15–45) to 0.2303 (95%CI 0.0208, 0.5733). Censoring did not change the survivor function or risk of outcome in the three sets of 5 days earlier indicated.

## Discussion

The study reports, for the first time in Ghana, indications for obstetric ICU admissions, their outcomes and the factors influencing the outcomes to aid continuous quality improvement in obstetric care among other benefits. Obstetric ICU admissions constituted 25.7% of all ICU

**Table 2. Factors influencing obstetric ICU admission outcomes in Cape Coast Teaching Hospital from 2010 to 2019.**

| &Variable | Crude Hazard Ratio (HR) | | *Adjusted Hazard Ratio (AHR) | |
|---|---|---|---|---|
| | HR (95% CI) | p-value | AHR (95% CI) | p-value |
| **Age (years)** | | | | |
| <25 | 1 | | 1 | |
| 25–35 | 1.67 (1.06, 2.62) | 0.025 | 1.41 (0.90, 2.22) | 0.132 |
| >35 | 2.18 (1.24, 3.84) | 0.007 | 1.88 (1.07, 3.31) | 0.028 |
| **Whether patient was intubated or not** | | | | |
| No intubation | 1 | | 1 | |
| Intubation | 6.70 (4.47, 10.05) | <0.001 | 6.37 (4.23, 9.57) | <0.001 |
| **Whether patient had surgery or not for the index pregnancy** | | | | |
| No surgery | 1 | | 1 | |
| surgery | 0.66 (0.45, 0.97) | 0.034 | 0.76 (0.51, 1.12) | 0.165 |
| **Occupation** | | | | |
| Formal | 1 | | 1 | |
| Informal | 1.97 (0.85, 4.53) | 0.110 | | |
| Unemployed | 1.41 (0.56, 3.51) | 0.466 | | |

*in the multivariable analysis giving the adjusted hazard ratio, the variables age, intubation and surgery were used

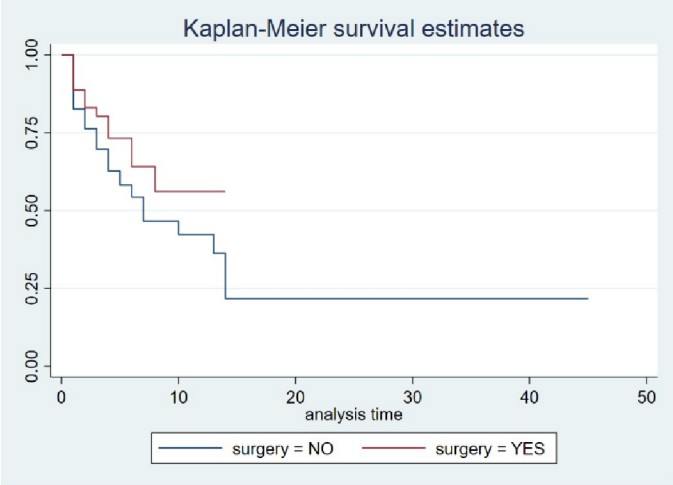

a: Survival estimates following obstetric ICU admission compared between participants who had surgery and those who had no surgery

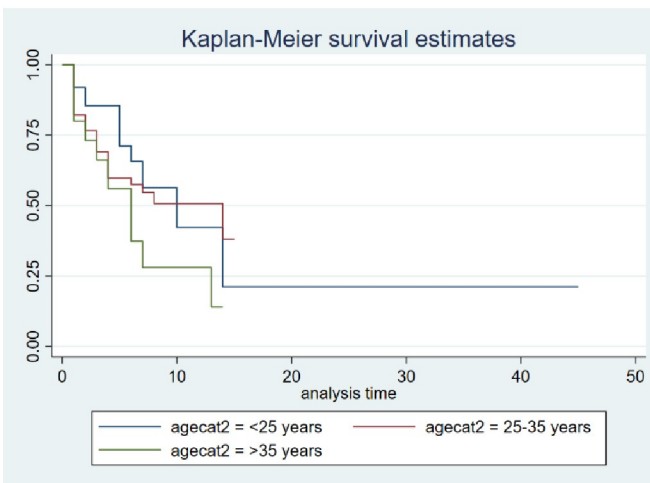

c: Survival estimates following obstetric ICU admission compared among different participant age groups

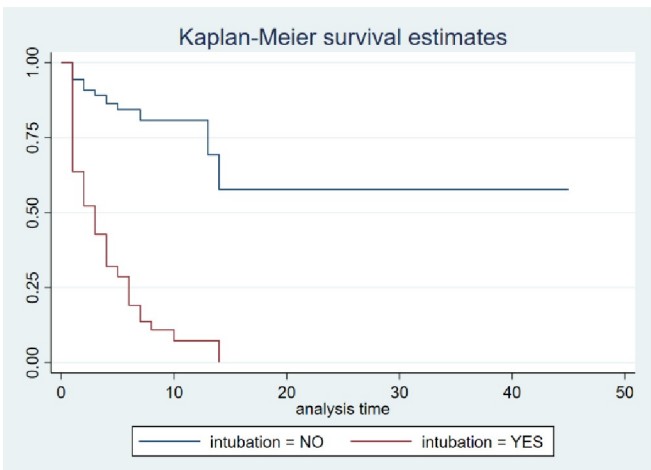

b: Survival estimates following obstetric ICU admission compared between participants who were intubated and those who were not

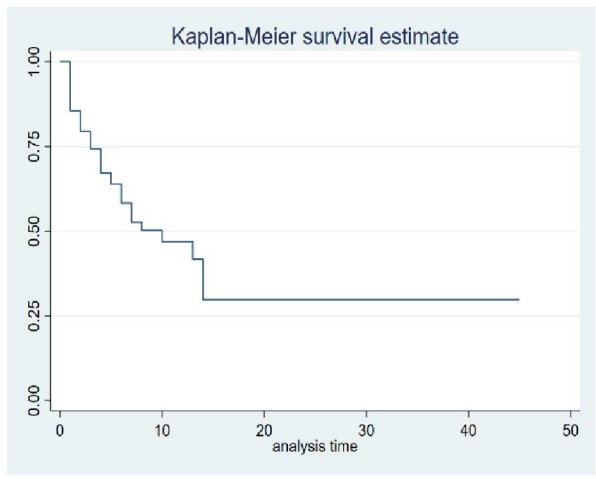

d: Overall survival estimates following ICU admission

**Fig 4. Survival estimates.**

admissions with an incidence of 14.7/1000 deliveries. The mortality rate following obstetric ICU admission was 26%. Higher age, need for mechanical ventilation and whether a patient had surgery for that particular obstetric experience were significant predictors of death. The global incidence of obstetric ICU admissions is reported as 0.4–16% of ICU admissions and 0.7–13.5 per 1000 deliveries, with lower rates of admission and mortality seen in developed countries [1, 2, 17].

The finding that obstetric cases accounted for 25.7% of all ICU admissions, is one of the highest ever recorded compared to previous reports [2, 10, 18–22] and may stem from increased numbers of referrals from other hospitals in CCTH's catchment area over time. This may be further compounded by the lack of a high-dependency unit or high-care area in the Obstetrics and Gynaecology department. It has been reported that having an HDU can reduce the ICU load by 5% [23]

Hypertensive disorders of pregnancy were the most prominent indication for ICU admission over the ten-year review period This is consistent with the global picture [1, 14], and the black racial group, primigravidae and increasing maternal age are known risk factors for their occurrence [24–26]. This finding indirectly reinforces other reports identifying HDP as the leading cause of maternal mortality [27–33].

Considering that HDP contributed to 70.4% of obstetric ICU admissions in our study with almost 70% having complications from hypertension, it is important to reiterate screening, prevention and early diagnosis of HDPs, particularly preeclampsia, to effect appropriate management and prevent severe morbidity that can lead to ICU admissions. Useful measures include public education on early antenatal booking and the danger signs of HDP, the development of standard operating protocols (SOPs) in identifying and managing at-risk pregnant women, training of healthcare workers in the SOPs and administering low-dose aspirin and calcium to at-risk women from the 12th week to the 36th week of pregnancy.

Haemorrhage and sepsis are also important causes of ICU admission across the world, though less prominent than HDP [6, 10–14, 19, 34]. Emergency preparedness at points of care units for pregnant women and on the labour wards is vital, as the first few minutes of hemorrhage represent a golden period within which impactful interventions can be made. A high index of suspicion for sepsis in at-risk women is needed, so that early diagnosis and treatment with appropriate antimicrobials can ensue. It is also important to institute antibiotics sensitivity pattern and stewardship to ensure a positive impact on the outcome of septic patients.

Increased numbers of referrals to CCTH since its status was upgraded from a regional hospital to a tertiary facility/teaching hospital in 2013 may well account for the increased percentages in haemorrhage, sepsis and medical causes as indications for obstetric ICU admission comparing the first half of the review period (2010–2014) to the second half (2015–2019). It is logical to presume increases in HDP-related referrals as well but a reduction is noted in 2015–2019 and this may be attributable to two reasons. First, in line with its upgrade, the number of Obstetrician-Gynaecologists in the hospital increased about four-fold. A postgraduate residency programme was also started within the same period. This increase in high and middle level manpower underpinned the management of less critical cases of HDP on the wards rather than in the ICU. The few ICU beds were therefore reserved for more critically-ill patients. Secondly, the management protocol for HDP in the hospital was reviewed in 2015. Among others, random blood sugar, bedside clotting time and evaluation of circulatory volume loss in severely-ill patients were regularly done and abnormalities corrected appropriately.

Although obstetric cases accounted for a relatively high proportion of all ICU admissions compared to other countries [18–22], the associated mortality rate of 26% is markedly lower than the 35%-54% reported in other African countries [2, 6, 10, 35]. However, it remains higher than rates reported in high-income countries including Australia and the Netherlands [11, 14, 36]. It is unclear why the proportion of obstetric ICU admissions that died in the present study is lower than that in other African countries where the health care system is unlikely to be different from that of Ghana. A comparative evaluation of the health system and access in these different countries would be needed to uncover possible reasons for the discrepancy. The lower mortality rates in high income countries, on the other hand, are likely to be the

result of early reporting to the hospital, better equipped and higher-level ICUs, expertise and the general health infrastructure present in those jurisdictions.

The study observed increased risks of ICU deaths for women who were at least 25 years old compared to those less than 25 years with the greatest risk in those above 35 years. This agrees with a study in Nigeria that reported higher age was significantly associated with maternal deaths following ICU admission [37]. This is expected in the category of women 35 years and above as this age group has greater risks of obstetric complications and possible underlying medical conditions [25, 38, 39] that may invariably predispose them to death. Although the outcome for those less than 25 years looked good, attention must be paid to the teenagers in that group as they carry increased risk of obstetric complications arising variously from bio-logic immaturity, poor antenatal clinic attendance, and several psycho-social stressors [40, 41]. Access to adolescent and sexual health education, contraception and schooling must be promoted in this group, especially in the Central Region, as teenage pregnancy has long been a noticeable challenge in the region [42].

Over the ten-year review period in the present study, 28.9% of obstetric ICU admissions had a need for mechanical ventilation and were intubated. This is much lower than the 90% and 95% reported in Nigeria and Malawi respectively [2, 10] but comparable to or higher than that observed in some high-income environments [11, 36]. The difference between our findings and that reported in other African countries [2, 10] could stem from the severity of morbidity at presentation. The need for intubation suggests severe or critical illness which may arise from late referrals or new developments complicating existing conditions. It is possible that while late referrals to CCTH may have been lower compared to the study sites in Nigeria and Malawi, they remain high compared to what pertains in developed countries with more efficient referral systems.

The current study found that intubation carried at least six times risk of an obstetric ICU death and agrees with an earlier report where mechanical ventilation was a statistically significant predictor of obstetric ICU deaths [37]. Delayed access to ventilator support could lead to irreversible hypoxic damage and further worsen the plight of patients who need it. Increased accessibility to ventilators is thus vital for improved outcomes. It must be noted that other studies did not report a need for mechanical ventilation as a predictor of obstetric ICU deaths or admission [23, 43].

Having had surgery in the index pregnancy was associated with a reduced risk of obstetric ICU death in the present study. With surgery essentially being life-saving, this finding is expected. However, we are unable to situate it in the bigger context of existing literature as no study was found that reported the relation between having surgery for a particular obstetric experience and survival following ICU admission. The said relation did not appear to be confounded by participants' age or a need for mechanical ventilation as cross-tabulation showed that the categories of the variables age and intubation had similar proportions of those who had or did not have surgery.

With at least a 5-fold increase in its contribution from 2010–2014 to 2015–2019 observed in the review, the medical causes of obstetric ICU admission draw grave attention, beyond pregnant and post-partum women to the general population, regarding the increasing burden of disorders such as the cardiovascular diseases in Ghana [44, 45]. Presumably, a number of these obstetric cases had pre-existing disease that may have been aggravated by pregnancy. Promotive health measures to reduce risk factors such as smoking, inordinate alcohol consumption and sedentary lifestyles among others must be vigorously pursued to stem the tide. It also emphasizes the need for specialized fields, such as maternal-fetal medicine, critical care medicine and nursing, obstetric anaesthesiology, cardiology and neonatology to meet this emerging challenge in the country.

## Study limitations

In addition to using data from only one site and its associated limited generalizability, there was insufficient data on patients' level of education, parity, gravidity and antenatal care attendance history. How these variables relate to the outcomes of mortality or survival following obstetric ICU admission could not be assessed and this is deemed a study limitation. Educational level has been reported to be significantly associated with obstetric ICU death [37]. Also, there was no data on some participants' dates of admission and discharge/death but these constituted less than 5% of the number included in the review and is not likely to challenge the validity of the study findings. Lastly, the data collection process did not distinguish between participants who were receiving care in-house in CCTH prior to ICU admission and those referred from other hospitals. The study is thus unable to report on the survival or hazard function of referred patients. However, with care delivered by specialists in CCTH, we posit that complications are likely to be detected early and interventions instituted for better survival of patients who were receiving care in-house prior to ICU admission.

## Conclusion and recommendations

Hypertensive disorders of pregnancy and haemorrhage have been the topmost indications for obstetric ICU admissions in CCTH over the past decade. Maternal age of at least 25 years and a need for mechanical ventilation carry increased risks of mortality following ICU admissions. The study draws attention to the need for screening and prevention of preeclampsia, antenatal education on HDP and obstetric haemorrhage, and early reporting/referral. The obstetricians may consider forging mentorship relationships with doctors at the lower facilities so that the latter can, at least, call for advice when faced with difficult cases. Furthermore, there is a dire need for an HDU in the Obstetrics and Gynaecology department to ease pressure on the ICU. A prospective study that takes into consideration all relevant study variables and conducted in multiple sites in the country is recommended to address the study limitations and better appreciate the wider scope of issues underlying obstetric ICU admission outcomes.

## Acknowledgments

We acknowledge the unknown participants whose data we used. We are also grateful to the hospital management for the permission to access patient data.

## Author Contributions

**Conceptualization:** Betty Anane-Fenin.

**Data curation:** Betty Anane-Fenin, Evans Kofi Agbeno, Joseph Osarfo, Douglas Aninng Opoku Anning, Abigail Serwaa Boateng, Sebastian Ken-Amoah, Anthony Ofori Amanfo, Leonard Derkyi-Kwarteng, Mohammed Mouhajer, Sarah Ama Amoo, Joycelyn Ashong, Ernestina Jeffery.

**Formal analysis:** Joseph Osarfo, Douglas Aninng Opoku Anning.

**Methodology:** Betty Anane-Fenin.

**Project administration:** Betty Anane-Fenin, Evans Kofi Agbeno.

**Supervision:** Betty Anane-Fenin, Evans Kofi Agbeno.

**Validation:** Betty Anane-Fenin, Evans Kofi Agbeno, Joseph Osarfo, Douglas Aninng Opoku Anning.

**Writing – original draft:** Betty Anane-Fenin.

**Writing – review & editing:** Betty Anane-Fenin, Evans Kofi Agbeno, Joseph Osarfo, Douglas
Aninng Opoku Anning, Abigail Serwaa Boateng, Sebastian Ken-Amoah, Anthony Ofori
Amanfo, Leonard Derkyi-Kwarteng, Mohammed Mouhajer, Sarah Ama Amoo, Joycelyn
Ashong, Ernestina Jeffery.

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
