## [Decision Letter · Decision Letter 0]

6 Sep 2021

PONE-D-21-15253A ten-year review of indications and outcomes of obstetric admissions to an intensive care unit in a low-resource countryPLOS ONE

Dear Dr. Anane-Fenin,

Thank you for submitting your manuscript to PLOS ONE. After careful consideration, we feel that it has merit but does not fully meet PLOS ONE’s publication criteria as it currently stands. Therefore, we invite you to submit a revised version of the manuscript that addresses the points raised during the review process.

We look forward to receiving your revised manuscript.

Kind regards,

Orvalho Augusto, MD, MPH

Academic Editor

PLOS ONE

2. In the ethics statement in the manuscript and in the online submission form, please provide additional information about the patient records used in your retrospective study, including: a) whether all data were fully anonymized before you accessed them; b) the date range (month and year) during which patients' medical records were accessed; c) the date range (month and year) during which patients whose medical records were selected for this study sought treatment. If the ethics committee waived the need for informed consent, or patients provided informed written consent to have data from their medical records used in research, please include this information.

3. You indicated that you had ethical approval for your study. In your Methods section, please ensure you have also stated whether you obtained consent from parents or guardians of the minors (< 18) included in the study or whether the research ethics committee or IRB specifically waived the need for their consent.

Additional Editor Comments (if provided):

This is an important report on the causes of ICU admission and mortality among obstetric patients in Ghana. The authors did a revision of the charts of the last 10 years in the context of limited personnel and infrastructure. However, there is a quite missed analysis opportunity.

Major issues:

- The authors apparently have data from each woman (with very few missing data) from ICU admission to death or discharge from the ICU. Why not perform survival analysis here? A Kaplan Meier plot would show with more granularity the mortality. That is more informative for action.

- The point above would have alerted the authors not to use the length of stay as a factor for mortality in table 3. Conduct an appropriate survival regression that uses both the death variable and length of stay as outcome (Cox proportional hazards model, or even discrete logistic regression).

- The authors have a longitudinal dataset. Why not check whether over time the pattern of mortality and its causes changed. Are the conditions killing in 2019 the same as in the very first years of 2010s? And did the mortality levels change (improve) over time?

- Please indicate whether are you using the ICD coding and what version are using.

Minor issues:

I. Abstract

There is a lack of aims in the background.

II. Background

No comments

III. Methods

Can you indicate whether are using ICD? And what version of ICD? And it would be good to indicate in the subsection data items the code.

Line 160 - “placental abruptio” should be “placental abruption”

Line 185 - Correct it to be: “The significance level was set at 5%, and 95% confidence intervals are reported”. Remember in order to reject a null hypothesis from a confidence interval the null must be out of the interval. Currently, this line does not mean that.

IV. Results

Line 193 - the 443 participants were not recruited. You did include them and among the 443 it is unclear whether some were included twice (on different pregnancies). So be precise in the language.

Lines 193 to 196 - Are the 1721 ICU admissions from the women delivering?

Line 204 - I cannot read this line on the PDF. I got it from the word document.

Table 3 - we need measure of association (odds-ratio, relative risk or hazard ratios) rather than just p-values. Please see the second major issue.

Table 4 - I cannot read table 4 in the PDF. Had to get it from the Word document. How the covariates for this analysis were selected? That process must be explained in the statistical section.

Figure 1, 2 - Couple of this:

Make on as 1a) and the 1b)

please make plain bar charts. Avoid these 3D features. And please add a y-axis.

Put clearer caption for these figures. This figure must stand out alone. As of now, “Indications for admission” or “category of hypertension” are insufficient.

Figure 3 is misleading

Subsection “Relationship between age and hypertension” - why age is categorized this way?

VI. Discussion

Please add a discussion for the limitations of this study.

Reviewers' comments:

Reviewer's Responses to Questions

**Comments to the Author**

1. Is the manuscript technically sound, and do the data support the conclusions?

Reviewer #1: Partly

2. Has the statistical analysis been performed appropriately and rigorously? 

Reviewer #1: Yes

3. Have the authors made all data underlying the findings in their manuscript fully available?

Reviewer #1: Yes

4. Is the manuscript presented in an intelligible fashion and written in standard English?

Reviewer #1: Yes

5. Review Comments to the Author

Reviewer #1: 1. Abstract:

The conclusion seems inconsistent with objective/purpose and available data from the study. The researchers should align their conclusions to the indications for admission

2. Plain English summary:

1) This manuscript seems not fitting to the PLOS ONE template. Please consider including plain English summary right after the abstract section for lay readers

3. Introduction:

1) Study objectives missing in the background section of the manuscript

4. Methodology

It’s very important the researchers make brief descriptions on how obstetric charts were retrieved, whether records were kept electronically or in manual archives, how the ICU data were linked to the obstetric information in the maternity chart including transfer notes

Was there any exclusion/ inclusion criteria to guide the data extraction? For instance, some cases didn’t have date of transfer or death in the ICU and wondering how this affected the total study sample and data collection process

Please clearly indicate whether the cases were transferred to ICU from the same hospital or referrals from other health facilities in the regions taken place. The researchers should present statistics on this data. This can be linked to the question above as how the accessed patient data on the recent obstetric experience particularly if the cases were referred from other health facilities. The researchers should explain these details in the data collection section

5. Results:

1) Page 14 - A total of 443 participants between the ages of 14 and 48 were recruited into the study. What was the rationale for including only 14-48 age bracket in this study?

2) Page 14 - There were 298 maternal mortalities, 115 obstetric deaths in ICU. What was the difference between maternal mortality and obstetric death? Please explain and use languages consistently.

3) Page 10 - Missing values for either date of admission or date of transfer/ death were approximately 3.4% (15 out of 443). If the date of transfer/death was missing were these excluded from analysis? Please confirm

4) Page 10 - About 47% of all participants (n=207) had undergone recent surgery 213 (caesarean section or laparotomy for uterine repair, postpartum hysterectomy or other 214 indications). The temporal reference in this statement lacks precision and very confusing to determine any association with the outcomes of interest. What is “recent”? Does this refer to the current obstetric experience?

5) What is the relevance of presenting Fig headings on page 10? Does this comply with the journal’s manuscript template? I found it odd to see the three disjoined headings in this section of the result.

6) A fundamental question was whether all sturdy participants were transferred from labour ward immediately after of during childbirth

7) While the sample size seems very low given the ten year period, it’s unfortunate the researchers didn’t show the pattern of ICU admission incidents and associated outcomes over time. I still urge them to discuss this trend as much as possible to help appreciate changes and to trigger health care planning accordingly

6. Discussion:

On page 13 – argument related to the 25.7% ICU admission “the highest ever”, which focused on the absence of adequate facility and geographic coverage doesn’t seem cogent with the indications and outcomes. Furthermore, the suggested expansion of ICU facilities personnel on page 14 should come under conclusion/recommendation and not as part of the discussion

Page 14- “Although the outcome for those less than or equal to 17 years were generally good…” The researchers should discuss their results based on data and in comparison with previous research findings rather than based on hypothetical statements. I suggest they should argue how and why their findings for this age group was different from their hypothesis and how literature confirms that

Page 15 – The researchers presented general reality about hemorrhage and sepsis as indications for ICU admission however they failed to discuss their own findings in this regard and compare to other similar literature. This is very important and can improve the quality of this paper

Are the patient factors confounded by other clinical condition or delays in seeking care?

7. Conclusion

As highlighted in the abstract section, the conclusion should also include significance of indications for obstetric ICU admission and how these could be addressed which is the central objective of the study

6. PLOS authors have the option to publish the peer review history of their article (what does this mean?). If published, this will include your full peer review and any attached files.

Reviewer #1: No

---

## [Author Response · Author response to Decision Letter 0]

23 Oct 2021

1. In the ethics statement in the manuscript and in the online submission form, please provide additional information about the patient records used in your retrospective study, including: a) whether all data were fully anonymized before you accessed them; b) the date range (month and year) during which patients' medical records were accessed; c) the date range (month and year) during which patients whose medical records were selected for this study sought treatment. If the ethics committee waived the need for informed consent, or patients provided informed written consent to have data from their medical records used in research, please include this information.

Response a: An ‘Ethical Considerations’ sub-section has been included in the manuscript (lines 236 to 242) and it addresses issues of anonymization of extracted data and informed consent. 

It reads as follows;

“Ethical clearance for this secondary data review was granted by the Ethical Review Committee of CCTH with reference number CCTHERC/EC/2020/080. All data were fully anonymized using assigned study identification numbers to assure confidentiality. The study was retrospective in approach and spanned a decade. Informed consent was not obtained as participants had either been long discharged or dead at the time of data extraction. This was made clear in the study proposal submitted for ethical review and it did not generate any queries. The hospital management also gave written permission to use patient data.”

Response b and c: There was no ‘selection’ per se as the review involved data on all obstetric patients who had been on admission at the ICU from 1st Jan 2010 to 31st Dec 2019. The section on ‘Data Collection’ has been thoroughly reviewed to address the suggestions made in lines 162 to 194 as follows:

“The review involved all obstetric ICU admissions from 2010 to 2019 and hence no sample size determination was required. The inclusion criterion was obstetric ICU admission (including those up to 42 days post-partum) for any period from 1st January 2010 to 31st December 2019. A participant was to be excluded if data could not be found on ≥40% of the planned variables.

From 3rd February, 2020 to 31st July, 2020, data on patient age, occupation, diagnosis, dates of admission and discharge, whether patient had surgery as part of that particular obstetric experience prior to ICU admission or required mechanical ventilation during admission at the ICU and the final outcome following admission were extracted mainly from archived scanned versions of patient folders, the admissions and discharges (A&D) register, the report and incident books at the ICU, the patient register at the theatre as well as patient records from the hospital’s electronic data registry. This was done for all obstetric patients meeting the inclusion criterion and none was excluded. Data extraction was done by two residents in the department who were trained for two days on the study objectives, definition of terminologies and the study protocols. A template designed by the first author was used for the extraction. For quality control, at least 70% of the data collected was cross-checked against primary sources for accuracy and consistency by three obstetricians who worked closely with the residents. Data on other relevant demographics such as level of education, parity and gravidity were poorly documented and could not be used. To give more context, the hospital used paper patient folders until 2017. In 2018, patient records went fully electronic as part of a nationwide implementation of electronic health record-keeping based on the international classification of disease (ICD) coding system and patient data could be accessed at all points of care in the hospital. Hard-paper patient folders were scanned and archived but a 100% completion may not have been achieved. Data collection for the review thus employed both scanned folders and the electronic registry among others as earlier indicated. Also, all points of care keep summary records of patient care and these records were used to triangulate data collected and to fill in missing information as much as possible.

In addition, total ICU admissions and deaths (obstetric and non-obstetric), total number of deliveries and the total number of maternal mortalities recorded in the hospital (in and outside the ICU) over the review period were also captured to contextually situate obstetric ICU admissions and deaths.”

2. You indicated that you had ethical approval for your study. In your Methods section, please ensure you have also stated whether you obtained consent from parents or guardians of the minors (< 18) included in the study or whether the research ethics committee or IRB specifically waived the need for their consent.

Response: Please see ‘Response a’ given above. The question of informed consent or assent was not deemed to have locus. This was made clear in the documents submitted for ethical review and approval was granted. 

Response:

We wish to make changes to the Data Availability statement. The change is to the effect that all data have been submitted as part of the manuscript. This change will be emphasized in our resubmission cover letter for subsequent update.

4. Major issues:

- The authors apparently have data from each woman (with very few missing data) from ICU admission to death or discharge from the ICU. Why not perform survival analysis here? A Kaplan Meier plot would show with more granularity the mortality. That is more informative for action.

- The point above would have alerted the authors not to use the length of stay as a factor for mortality in table 3. Conduct an appropriate survival regression that uses both the death variable and length of stay as outcome (Cox proportional hazards model, or even discrete logistic regression).

- The authors have a longitudinal dataset. Why not check whether over time the pattern of mortality and its causes changed. Are the conditions killing in 2019 the same as in the very first years of 2010s? And did the mortality levels change (improve) over time?

- Please indicate whether are you using the ICD coding and what version are using.

Response: 

The suggestions raised are profoundly acknowledged and have been heeded to. We re-approached the data as time-to-event data and performed survival analysis to generate Hazard Ratios with 95% confidence intervals and p-values for mortality as an outcome of obstetric ICU admission (see Table 2 under Results).

We have also included some dimensions of comparison of obstetric ICU admission indications and mortality between the periods 2010-2014 and 2015-2019 and these have been reported (see Fig 3 under Results). The findings have also been appropriately discussed from lines 381 to 394 in the ‘Discussion’.

5. Abstract

There is a lack of aims in the background.

Response: 

The background has been reviewed to capture the concern raised. It now reads as follows:

“Obstetric intensive care unit admission (ICU) suggests severe morbidity. However, there is no available data on the subject in Ghana. This retrospective review was conducted to determine the indications for obstetric ICU admission, their outcomes and factors influencing these outcomes to aid continuous quality improvement in obstetric care.’’ 

6. Methods

Can you indicate whether are using ICD? And what version of ICD? And it would be good to indicate in the subsection data items the code.

Response:

No. The morbidity classification was not based on ICD. It was arbitrary as the hospital only started using an ICD-based electronic record system in 2018. This has been stated under ‘Description of diagnoses categories used’ under Data Collection.

7. Line 160 - “placental abruptio” should be “placental abruption”

Response: This correction has been done (lines 202 and 203)

8. Line 185 - Correct it to be: “The significance level was set at 5%, and 95% confidence intervals are reported”. Remember in order to reject a null hypothesis from a confidence interval the null must be out of the interval. Currently, this line does not mean that.

Response: The correction has been made under ‘Data analysis’ (lines 229 and 230)

9. Line 193 - the 443 participants were not recruited. You did include them and among the 443 it is unclear whether some were included twice (on different pregnancies). So be precise in the language.

Response: This was an oversight on our part. It has been corrected to now read (under Background and clinical characteristics of study participants under Results from lines 247 to 250) as;

“Four hundred and forty-three (443) clients were admitted to the ICU over the defined period and they all contributed data to the review. None of the participants had a repeat ICU admission on a different pregnancy. Each participant thus contributed data only once to the review……..”

10.Lines 193 to 196 - Are the 1721 ICU admissions from the women delivering?

Response: 1721 was the number of all ICU admissions (obstetric plus non-obstetric over the ten-year review period). It was specifically captured to help situate obstetric ICU admissions in context. This has been explained under Data Collection.

11.Line 204 - I cannot read this line on the PDF. I got it from the word document.

Response: We apologize for this but we are not sure where the problem is coming from. The PDF is supposed to be built from the word document submitted. I do hope it will be picked up by the editorial/production team on acceptance for publication.

12. Table 3 - we need measure of association (odds-ratio, relative risk or hazard ratios) rather than just p-values. Please see the second major issue.

Response: Table 3 NO LONGER exists. It has been replaced with the present Table 2 which gives crude and adjusted hazard ratios following Cox regression analysis.

13. Table 4 - I cannot read table 4 in the PDF. Had to get it from the Word document. How the covariates for this analysis were selected? That process must be explained in the statistical section.

Response: The question has been addressed as follows in the Data Analysis section (lines 229 to 232):

“…The significance level was set at 5%, and 95% confidence intervals are reported. Variables with significant p-values (p<0.05 at 95% confidence interval) in univariate analysis were entered into a multivariate regression model for adjusted hazard ratios…”

14. Figure 1, 2 - Couple of this:

Make on as 1a) and the 1b)

please make plain bar charts. Avoid these 3D features. And please add a y-axis.

Put clearer caption for these figures. This figure must stand out alone. As of now, “Indications for admission” or “category of hypertension” are insufficient.

Response: 

These concerns have been addressed. Please see the current Figs 1-3 under Results.

On the matter of making the first two figures 1a and 1b, we feel it may contravene the journal’s requirements as it prefers numbering figures numerically in increasing order.

15. Figure 3 is misleading

Response: The ‘misleading’ Fig.3 has been expunged. The new Fig. 3 is entirely different and compares indications for obstetric ICU admission over the periods 2010-2014 and 2015-2019

16. Subsection “Relationship between age and hypertension” - why age is categorized this way?

Response: Age has been re-categorized to <25, 25-35 and >35 to reflect known age risks for hypertensive diseases of pregnancy. The particular subsection no longer exists on its own. It is now captured under ‘Factors influencing outcomes following obstetric ICU admission’

17. Discussion

Please add a discussion for the limitations of this study.

Response: This has been done. It is the last paragraph of the Discussion from lines 461 to 474 and reads as;

“In addition to using data from only one site and its associated limited generalizability, there was absence of data on patients’ level of education, parity, gravidity and antenatal care attendance history. How these variables relate to the outcomes of mortality or survival following obstetric ICU admission could not be assessed and this is deemed a study limitation. Educational level has been reported to be significantly associated with obstetric ICU death [37]. Also, there was no data on some participants’ dates of admission and discharge/death but these constituted less than 5% of the number included in the review and is not likely to challenge the validity of the study findings. Lastly, the data collection process did not distinguish between participants who were receiving care in-house in CCTH prior to ICU admission and those referred from other hospitals. The study is thus unable to report on the survival or hazard function of referred patients. However, with care delivered by specialists in CCTH, we posit that complications are likely to be detected early and interventions instituted for better survival of patients who were receiving care in-house prior to ICU admission.”

RESPONSE TO REVIEWER COMMENTS

1. Abstract:

 The conclusion seems inconsistent with objective/purpose and available data from the study. The researchers should align their conclusions to the indications for admission

Response: We agree with the reviewer on this and revisions have been made. It now reads as;

“Hypertension, haemorrhage and sepsis are the leading indications for obstetric ICU admissions. Thus, preeclampsia screening and prevention, as well as antenatal education on the danger signs of pregnancy can minimize obstetric complications. The establishment of an obstetric HDU in CCTH and the strengthening of communication between specialists and the healthcare providers in the lower facilities, are also essential for improved pregnancy outcomes. Further studies are needed to better appreciate the wider issues underlying obstetric ICU admission outcomes.’’ 

2. Plain English summary:

This manuscript seems not fitting to the PLOS ONE template. Please consider including plain English summary right after the abstract section for lay readers

Response: We did not find Plain English summary in the journal submission guidelines. Nonetheless, we have included a 143-word summary in plain language in the section below the abstract. It reads:

“This was a review of the reasons for admitting severely-ill pregnant women and women who had delivered within the past 42 days to the intensive care unit (ICU), the admission outcomes and risk factors associated with ICU mortality in a tertiary hospital in a low-resource country. High blood pressure and its complications, bleeding and severe infections were observed as the three most significant reasons for ICU admissions in decreasing order of significance. Pre-existing medical conditions and those arising as a result of, or aggravated by pregnancy; obstructed labour and post-operative monitoring were the other reasons for ICU admission over the study period. Overall, 26% of the admitted patients died at the ICU and maternal age of at least 25 years and the need for intubation were identified as risk factors for ICU deaths. Attention must be paid to high blood pressure during pregnancy.”

3. Introduction:

Study objectives missing in the background section of the manuscript

Response: The background has been reviewed and the relevant section now reads as;

“…it is pertinent to have an overarching knowledge of obstetric ICU admission indications, trends and outcomes that inure to the benefit of all stakeholders including clinicians, managers and those in the health planning and policy space.

This ten-year review was conducted to obtain an overview of indications for obstetric ICU admission, outcomes and factors influencing these outcomes. It is expected to provide evidence-based insights for much desired continuous quality improvement, especially for the critically ill, in the Cape Coast Teaching Hospital and other tertiary facilities in Ghana and sub-Saharan Africa.”

4. Methodology

It’s very important the researchers make brief descriptions on how obstetric charts were retrieved, whether records were kept electronically or in manual archives, how the ICU data were linked to the obstetric information in the maternity chart including transfer notes

Response: 

We agree we were not thorough in our description of accessed records the first time. The Data Collection section under Materials and Methods now has a detailed description of the nature of primary records and how they were accessed. The relevant sections read as follows (lines 167 to 190):

“...From 3rd February, 2020 to 31st July, 2020, data on patient age, occupation, diagnosis, dates of admission and discharge, whether patient had surgery as part of that particular obstetric experience prior to ICU admission or required mechanical ventilation during admission at the ICU and the final outcome following admission were extracted mainly from archived scanned versions of patient folders, the admissions and discharges (A&D) register, the report and incident books at the ICU, the patient register at the theatre as well as patient records from the hospital’s electronic data registry……… To give more context, the hospital used paper patient folders until 2017. In 2018, patient records went fully electronic as part of a nationwide implementation of electronic health record-keeping based on the international classification of disease (ICD) coding system and patient data could be accessed at all points of care in the hospital. Hard-paper patient folders were scanned and archived but a 100% completion may not have been achieved. Data collection for the review thus employed both scanned folders and the electronic registry among others as earlier indicated. Also, all points of care keep summary records of patient care and these records were used to triangulate data collected and to fill in missing information as much as possible...”

Was there any exclusion/ inclusion criteria to guide the data extraction? For instance, some cases didn’t have date of transfer or death in the ICU and wondering how this affected the total study sample and data collection process

Response: Inclusion/exclusion criteria have been defined as below (under Data Collection from lines 162 to 166). There was no exclusion as none met the exclusion criteria.

“The review involved all obstetric ICU admissions from 2010 to 2019 and hence no sample size determination was required. The inclusion criterion was obstetric ICU admission (including those up to 42 days post-partum) for any period from 1st January 2010 to 31st December 2019. A participant was to be excluded if data could not be found on ≥40% of the planned variables.”

The matter of those who had missing dates of admission/discharge/death has been discussed as a study limitation from lines 466 to 468. The relevant section reads as:

“…Also, there was no data on some participants’ dates of admission and discharge/death but these constituted less than 5% of the number included in the review and is not likely to challenge the validity of the study findings…”

Please clearly indicate whether the cases were transferred to ICU from the same hospital or referrals from other health facilities in the regions taken place. The researchers should present statistics on this data. This can be linked to the question above as how the accessed patient data on the recent obstetric experience particularly if the cases were referred from other health facilities. The researchers should explain these details in the data collection section

Response:

Case flow to the ICU has been described and emphasized under Study Site Description. The review was carried out in a teaching hospital which receives referrals from a wide catchment area far beyond the Central Region where it is located. Certainly, some of the participants in the review were such referred cases. However, the data collection process did not distinguish between referred obstetric patients and those who were receiving care in-house prior to ICU admission. We are thus unable to present statistics on the different groups (referred and in-house). This has been captured under the study limitation (from lines 468 to 474) and reads as below:

“Lastly, the data collection process did not distinguish between participants who were receiving care in-house in CCTH prior to ICU admission and those referred from other hospitals. The study is thus unable to report on the survival or hazard function of referred patients. However, with care delivered by specialists in CCTH, we posit that complications are likely to be detected early and interventions instituted for better survival of patients who were receiving care in-house prior to ICU admission.”

5. Results:

Page 14 - A total of 443 participants between the ages of 14 and 48 were recruited into the study. What was the rationale for including only 14-48 age bracket in this study?

Response: The statement above has been revised (lines 247 and 248) and now reads as below;

“Four hundred and forty-three (443) obstetric patients were admitted to the ICU over the defined study period and they all contributed data to the review.”

The age bracket indicated (14-48 years) was not deliberately chosen. This work was a review of secondary data and this age bracket is the range of ages of the 443 obstetric patients included in the review.

Page 14 - There were 298 maternal mortalities, 115 obstetric deaths in ICU. What was the difference between maternal mortality and obstetric death? Please explain and use languages consistently.

Response: Thank you for drawing our attention to the potential ambiguity here. 

As part of data collection, we also recorded the total number of maternal mortalities that occurred in the hospital (298) over the review period of 10 years. This total number entails both maternal deaths occurring in the ICU and those occurring outside the ICU in other units of the department of Obstetrics and Gynaecology. This has been made clear under ‘Data Collection’ (lines 191 to 194) and ‘Results’ (lines 254 to 256) as shown below;

“In addition, total ICU admissions and deaths (from all departments in CCTH) and the total number of maternal mortalities recorded in the hospital (in and outside the ICU) over the review period were also captured to contextually situate obstetric ICU admissions and deaths.”

“About a quarter of the obstetric ICU admissions died (26%, 115/443) and this number made up nearly two-fifths (38.6%, 115/298) of total maternal mortalities in the hospital over the review period...”

Page 10 - Missing values for either date of admission or date of transfer/ death were approximately 3.4% (15 out of 443). If the date of transfer/death was missing were these excluded from analysis? Please confirm.

Response: They were included in the analysis on two grounds. First, they had full data contribution with regard to other variables aside date of admission and date of discharge/death and thus did not meet the exclusion criteria on no data on ≥40% of variables assessed. Secondly, this is not expected to matter significantly as the quantum of missing data is less than 5%. This point of view is expressed as part of the study limitations in lines 466 to 468 as follows;

“...Also, there was no data on some participants’ dates of admission and discharge/death but these constituted less than 5% of the number included in the review and is not likely to challenge the validity of the study findings…”

Page 10 - About 47% of all participants (n=207) had undergone recent surgery 213 (caesarean section or laparotomy for uterine repair, postpartum hysterectomy or other 214 indications). The temporal reference in this statement lacks precision and very confusing to determine any association with the outcomes of interest. What is “recent”? Does this refer to the current obstetric experience?

Response: Thank you for drawing attention to this spot of ambiguity. The temporal context of ‘surgery’ has been better defined in the second paragraph under ‘Data Collection’ in lines 168 and 170. The sentence in question now reads;

“…..whether patient had a caesarean section/ hysterotomy, postpartum laparotomy for exploration, uterine repair or postpartum hysterectomy in the index pregnancy prior to ICU admission…….”

What is the relevance of presenting Fig headings on page 10? Does this comply with the journal’s manuscript template? I found it odd to see the three disjoined headings in this section of the result.

Response: This was an oversight and has been appropriately addressed

A fundamental question was whether all sturdy participants were transferred from labour ward immediately after of during childbirth

Response: As mentioned earlier, aside those patients who were probably receiving care in CCTH prior to ICU admission, the participants most certainly included referrals from other facilities in CCTH’s catchment area but our study failed to distinguish between them. More importantly, we did not capture data on how long after delivery a patient was kept on the ward (in the case of those seen at CCTH before ICU admission) or in a particular hospital (for referred cases) before ICU admission. These form part of the ‘relevant data’ further studies need to address as recommended in the Conclusion.

While the sample size seems very low given the ten year period, it’s unfortunate the researchers didn’t show the pattern of ICU admission incidents and associated outcomes over time. I still urge them to discuss this trend as much as possible to help appreciate changes and to trigger health care planning accordingly

Response: This has been done now. Please see ‘Description of trends in indicators for obstetric ICU admission comparing 2010-2014 and 2015-2019’ under Results (lines 285 to 296). Please see also the accompanying Fig.3 

Discussion:

On page 13 – argument related to the 25.7% ICU admission “the highest ever”, which focused on the absence of adequate facility and geographic coverage doesn’t seem cogent with the indications and outcomes. Furthermore, the suggested expansion of ICU facilities personnel on page 14 should come under conclusion/recommendation and not as part of the discussion

Page 14- “Although the outcome for those less than or equal to 17 years were generally good…” The researchers should discuss their results based on data and in comparison with previous research findings rather than based on hypothetical statements. I suggest they should argue how and why their findings for this age group was different from their hypothesis and how literature confirms that

Page 15 – The researchers presented general reality about hemorrhage and sepsis as indications for ICU admission however they failed to discuss their own findings in this regard and compare to other similar literature. This is very important and can improve the quality of this paper

Are the patient factors confounded by other clinical condition or delays in seeking care?

 Response:

The ‘Discussion’ has been markedly revised to address all the concerns raised. 

On the issue of good outcome for those less than 25 years (age has been re-categorized to reflect known risks for hypertensive disorders of pregnancy and this group was the reference for comparison. See Table 2). We have discussed the greater risk of deaths following ICU admission in older women. However, the age group (less than 25 years) includes teenagers and we feel the need to discuss them as well in light of a high burden of teenage pregnancy reported in the locality in another study. 

As mentioned in a response above, we did not capture data that can enable us make pronouncements on potential delays in seeking care. We are thus unable to speak confidently towards it. It makes sense to presume delays in seeking care will predispose to mortality and subtle directions to this end have been made in the Discussion.

Conclusion

As highlighted in the abstract section, the conclusion should also include significance of indications for obstetric ICU admission and how these could be addressed which is the central objective of the study

Response: The ‘Conclusion’ has been revised to address these concerns and now reads as;

“Hypertensive disorders of pregnancy and haemorrhage have been the topmost indications for obstetric ICU admissions in CCTH over the past decade. Maternal age greater than 25 years and a need for mechanical ventilation carry increased risks of mortality following ICU admissions. The study draws attention to the need for screening and prevention of preeclampsia, antenatal education on HDP and obstetric haemorrhage, and early reporting/referral. The obstetricians may consider forging mentorship relationships with doctors at the lower facilities so that the latter can, at least, call for advice when faced with difficult cases. Furthermore, there is a dire need for an HDU in the Obstetrics and Gynaecology department to ease pressure on the ICU. A prospective study that takes into consideration all relevant study variables and conducted in multiple sites in the country is recommended to address the study limitations and better appreciate the wider scope of issues underlying obstetric ICU admission outcomes.”

---

## [Decision Letter · Decision Letter 1]

1 Dec 2021

PONE-D-21-15253R1A ten-year review of indications and outcomes of obstetric admissions to an intensive care unit in a low-resource countryPLOS ONE

Dear Dr. Anane-Fenin,

Thank you for submitting your manuscript to PLOS ONE. After careful consideration, we feel that it has merit but does not fully meet PLOS ONE’s publication criteria as it currently stands. Therefore, we invite you to submit a revised version of the manuscript that addresses the points raised during the review process.

We look forward to receiving your revised manuscript.

Kind regards,

Orvalho Augusto, MD, MPH

Academic Editor

PLOS ONE

Journal Requirements:

Additional Editor Comments (if provided):

This is the second revision of this manuscript. The is authors present a much-needed review of intensive care among obstetric patients from a Teaching Hospital in a West African country. The data collection in itself shows how this exercise, although hard, can be done somewhere else to help to set priorities and hopefully, proper actions can happen.

The authors responded fully to the reviewers' comments and did, as requested, further analysis. I appreciate that.

Few more issues:

1. It would be good to have the abstract separated as introduction, methods, results and conclusion

2. Typos

a. Line 161 - No need for the “:”

b. Line 222 - No need for the “:”

c. Line 223 - please do not write STATA It is Stata (see the official Stata documentation).

d. Line 273 - No need for the “/”

e. Line 293 - It should be “Medical” not “Med9cal

3. Please correct from multivariate to multivariable in the whole document

4. In the “Factors influencing outcomes following obstetric ICU admission” subsection, please:

a. add descriptives of the length of stay. At least min/max and the mean.

b. Describe the overall survival as well. Example: that ¼ of the patients died in XX days (Stata sts list command should offer that). In the same line you can extract the median survival

5. Table 2 - please add the reference categories. For example, for age we need the “below 25” and as HR we would put “1” or “Reference”.

6. Figures 4, 5 and 6 could be placed in a single figure 4 with a, b and c. Furthermore:

a. Add an overall survival curve so you would have a), b), c) and d)

b. Report the risk set per each 5 day

c. Please censor time at 15 or 20th day.

7. Line 326 it mentions a supplementary file. I did not see such a file.

Reviewers' comments:

Reviewer's Responses to Questions

**Comments to the Author**

1. If the authors have adequately addressed your comments raised in a previous round of review and you feel that this manuscript is now acceptable for publication, you may indicate that here to bypass the “Comments to the Author” section, enter your conflict of interest statement in the “Confidential to Editor” section, and submit your "Accept" recommendation.

Reviewer #1: All comments have been addressed

2. Is the manuscript technically sound, and do the data support the conclusions?

Reviewer #1: Yes

3. Has the statistical analysis been performed appropriately and rigorously? 

Reviewer #1: Yes

4. Have the authors made all data underlying the findings in their manuscript fully available?

Reviewer #1: Yes

5. Is the manuscript presented in an intelligible fashion and written in standard English?

Reviewer #1: Yes

6. Review Comments to the Author

Reviewer #1: The authors have addressed the review comments adequately and I have no further comments. I recommend that this manuscript should be considered for publication.

7. PLOS authors have the option to publish the peer review history of their article (what does this mean?). If published, this will include your full peer review and any attached files.

Reviewer #1: **Yes: **Alemayehu Gebremariam Agena

---

## [Author Response · Author response to Decision Letter 1]

10 Dec 2021

1. It would be good to have the abstract separated as introduction, methods, results and conclusion

Response: The abstract has now been structured.

2. Typos

a. Line 161 - No need for the “:”

b. Line 222 - No need for the “:”

c. Line 223 - please do not write STATA It is Stata (see the official Stata documentation).

d. Line 273 - No need for the “/”

e. Line 293 - It should be “Medical” not “Med9cal

Response:

- ‘:’ has been deleted in line 166

- ‘:’ has been deleted in line 227

- STATA has been changed to Stata in line 228

- ‘/’ deleted in line 272

- Spelling of medical corrected in line 298

3. Please correct from multivariate to multivariable in the whole document

Response: ‘Multivariate’ have been corrected to ‘multivariable’ in lines 236 and 319

4. In the “Factors influencing outcomes following obstetric ICU admission” subsection, please:

a. add descriptives of the length of stay. At least min/max and the mean.

Response:

The length of stay, with the minimum and the maximum number of days were already provided under Background and Clinical Characteristics of participants in the Results section in lines 268 to 270. They have now been highlighted in red. 

The mean duration of stay has been included in line 270 and also in Table 1.

b. Describe the overall survival as well. Example: that ¼ of the patients died in XX days (Stata sts list command should offer that). In the same line you can extract the median survival

Response:

These have been provided in lines 307 to 310 as ‘About 14.5% (62/428) of study participants died on day 1 of ICU admission, 4.4% (19/428) on day 2 and 2.3% (10/428) on day 3. More than half of the ICU obstetric deaths (53.9%, 62/115) occurred within 24 hours of admission, 16.5% (19/115) on day 2 and 8.7% (10/115) on day 3. The median survival time is 10 days (95% CI 6,14). 

5. Table 2 - please add the reference categories. For example, for age we need the “below 25” and as HR we would put “1” or “Reference”.

Response: Table 2 has been revised.

6a. Add an overall survival curve so you would have a), b), c) and d)

Response:

The overall survival curve has been provided as the 4th graph in Figure 4.

All the survival curves have been put under one file and Figure 4 named ‘4a’, Figure 5 named ‘4b’, Figure 6 named ‘4c’, and the overall survival estimates named ‘4d’. 

6b. Report the risk set per each 5 day

6c. Please censor time at 15 or 20th day.

Response

If 6b and 6c are sub-parts of the same instruction, as we think, then they have been included in lines 334 to 341 as ‘Within the first 5 days, the survivor function was 0.7664 (95%CI 0.7233, 0.8036). In the second 5 days (i.e. days 5-9), survivor function was 0.6282 (95%CI 0.5415, 0.7030). In the third five days (i.e. 10-14), the survivor function was 0.4607 (95%CI 0.3047, 0.6032) and this particular survivor function remained constant over the next grouped days (15-45). Censoring time at day 15, the survivor functions for the first, second and third set of 5 days remained as reported above. However, the survivor function changed for the last set of days (15-45) to 0.2303 (95%CI 0.0208, 0.5733). Censoring did not change the survivor function or risk of outcome in the three sets of 5 days earlier indicated.

7. Line 326 it mentions a supplementary file. I did not see such a file.

Response:

We apologize for this mishap. The line has been deleted because all relevant files have already been provided.

---

## [Editor Report · Decision Letter 2]

15 Dec 2021

A ten-year review of indications and outcomes of obstetric admissions to an intensive care unit in a low-resource country

PONE-D-21-15253R2

Dear Dr. Anane-Fenin,

We’re pleased to inform you that your manuscript has been judged scientifically suitable for publication and will be formally accepted for publication once it meets all outstanding technical requirements.

Kind regards,

Orvalho Augusto, MD, MPH

Academic Editor

PLOS ONE

Additional Editor Comments (optional):

Just one small comment:

For the figures 4a), 4b), 4c) and 4d) can you censor the time at 20? If you cannot, OK to keep the plots as they are now.
---

## [Editor Report · Acceptance letter]

19 Dec 2021

PONE-D-21-15253R2 

A ten-year review of indications and outcomes of obstetric admissions to an intensive care unit in a low-resource country 

Dear Dr. Anane-Fenin:

I'm pleased to inform you that your manuscript has been deemed suitable for publication in PLOS ONE. Congratulations! Your manuscript is now with our production department. 

Kind regards, 

on behalf of

Dr. Orvalho Augusto 

Academic Editor

PLOS ONE